# Evolution of Protein Functional Annotation: Text Mining Study

**DOI:** 10.3390/jpm12030479

**Published:** 2022-03-16

**Authors:** Ekaterina V. Ilgisonis, Pavel V. Pogodin, Olga I. Kiseleva, Svetlana N. Tarbeeva, Elena A. Ponomarenko

**Affiliations:** Institute of Biomedical Chemistry, Pogodinskaya Street, 10, 119121 Moscow, Russia; pogodinpv@gmail.com (P.V.P.); olly.kiseleva@gmail.com (O.I.K.); tarbeevasn@gmail.com (S.N.T.); 2463731@gmail.com (E.A.P.)

**Keywords:** protein function, uPE1 proteins, missing proteins, neXtProt, text-mining, neXt-MP50, neXtCP-50, CHPP, Human Proteome Project

## Abstract

Within the Human Proteome Project initiative framework for creating functional annotations of uPE1 proteins, the neXt-CP50 Challenge was launched in 2018. In analogy with the missing-protein challenge, each command deciphers the functional features of the proteins in the chromosome-centric mode. However, the neXt-CP50 Challenge is more complicated than the missing-protein challenge: the approaches and methods for solving the problem are clear, but neither the concept of protein function nor specific experimental and/or bioinformatics protocols have been standardized to address it. We proposed using a retrospective analysis of the key HPP repository, the neXtProt database, to identify the most frequently used experimental and bioinformatic methods for analyzing protein functions, and the dynamics of accumulation of functional annotations. It has been shown that the dynamics of the increase in the number of proteins with known functions are greater than the progress made in the experimental confirmation of the existence of questionable proteins in the framework of the missing-protein challenge. At the same time, the functional annotation is based on the guilty-by-association postulate, according to which, based on large-scale experiments on API-MS and Y2H, proteins with unknown functions are most likely mapped through “handshakes” to biochemical processes.

## 1. Introduction

A clear understanding of a protein function is crucial for any systematic investigation of biological systems. Global DNA projects (Human Genome, 1KGenome, etc.) with technological advances in high-throughput sequencing facilitated the explosion of sequencing data, which, unfortunately, do not scalably translate into information at the proteome level.

The structures and sizes of the information space of genes and proteins differ significantly. Twenty thousand genes turn into millions of protein species, resulting from alternative splicing and carrying a unique profile of point mutations and post-translational modifications. These aberrations [1,2] outside of the genome’s control, make us who we are by fine-tuning the protein functionality.

Through the efforts of the participants in the Human Proteome Project, some progress has been made over 10 years of hard work: to date, we have obtained experimental confirmation of the existence of protein products of 18,354 human genes [3,4,5]. However, significant success in the inventory of the proteome does not answer the fundamental question: what is the role of each of the protein gears integrated into the vast and heterogeneous proteomic mechanism? 

According to the present proteomic knowledge, most protein entries are functionally annotated in automatic mode [6]. In particular, the majority of Gene Ontology (GO) annotations are automatically imported from various proteomic repositories without manual curation. Protein products are often assigned with functions, but it is not always obvious why a particular function was attributed to a certain protein [6].

The significant laboriousness, high cost, and complexity of experiments on the validation of protein functions [7] compared to bioinformatic approaches are reflected in the number of publications describing the functional properties of proteins. Most of the published data are based on predictions of functional properties of proteins, which may be inaccurate due to incomplete data or wrong interpretations [8]. Targeted experiments focused on unraveling the function of certain proteins are quite rare. Generally, the function is established based on bioinformatic interpretation of isolated experiments.

The complexity of functional annotation lies in the absence of a clear interpretation of the “function” and generally accepted standards for its definition. In a general sense, the function is everything that happens to or through a protein [9]. 

In practice, experimental evidence is substituted with computational inference through data-intensive methods, based on the ambiguous expansion of the relatively small number of experimentally validated functions to large sets of uncharacterized proteins. One of the most popular tools for exploring genes and proteins functions is Gene Ontology (GO) [10]. Gene Ontology attributes proteins with terms regarding their molecular function, involvement in biological processes, and subcellular localization. This platform provides the opportunity to distinguish functions between proteoforms encoded by the same gene and considers the function of a gene as the function of its products [11]. 

The accumulation of data on the proteome heterogeneity [12,13] and the functional differences between proteoforms [14,15] pushes the scientific community to rethink the distinction between the functions of a gene and a protein encoded by this gene.

We were intrigued by the understanding of protein function and used the neXtProt database [16] to investigate the evolution of this term. The neXtProt database is one of the most reliable databases regarding human proteins [16,17]. The majority of Gene Ontology (GO) annotations are automatically imported from various sources. In this paper, we analyzed the terms describing protein functions used in neXtProt and monitored how and why the “profile” of the functional diversity of the human proteome has changed in recent years. The “look into the past” of functional annotation allowed us to evaluate the readiness of the proteomic community for the transition from a description of the functions of genes to a description of the functions of specific proteoforms [18].

Overall, 20,239 protein-coding genes have been predicted from the analysis of the human genome (neXtProt release 18 February 2021), and about 10% of them are still lacking functional annotation, either predicted by bioinformatics tools or captured from experimental reports. Despite technological progress, the pace of human protein characterization studies is still slow. It could be accelerated by better integrating existing knowledge resources and by initiating large collaborative projects involving specialists from different fields of biology. Traditionally, gene/protein functions are first identified by in vitro and in vivo experiments and recorded in biological databases via literature-based curation. However, wet-lab investigations and manual curation efforts are cumbersome and time-consuming. Thus, they cannot resolve the knowledge gap produced due to the continuous growth of biological sequence data [1]. Therefore, accurate computational methods have been sought to annotate the functions of proteins automatically.

The existing computational instruments for protein function annotation are mostly based on the following data and approaches [8]: Sequence and structural analysis;Subcellular localization(s);Protein–protein interactions (Guilty by Association);Expression and coexpression;Phenotypes and diseases;Text mining (including GO annotation).

Protein function prediction from sequence using the Gene Ontology classification is helpful in many biological problems. One of the common approaches is “Guilty by Association” (GAS) on STRING, used to predict protein function by exploiting protein–protein interaction networks without knowledge about sequence similarity. The assumption is that whenever a protein interacts with other proteins, it is part of the same biological process and is located in the same cellular compartment. The Guilty by Association (GAS) on STRING tool predicts protein function by exploiting protein–protein interaction networks without sequence similarity and is used in many existing function prediction programs, e.g., NetGO and MetaGO.

Speaking about ready-to-use instruments for protein function prediction, it is necessary to mention the following: Numerous sequence-based function prediction methods are based on structural motifs possibly associated with a known biochemical function [19]. Today, competing state-of-the-art sequence-similarity-based prediction methods for homology search combined with deep convolutional neural network models (such as DeepGOPlus [20], GeneMANIA, deepNF, and clusDCA [21]) scan the sequence for motifs that are predictive for protein functions and combine this with functions of similar proteins with protein annotation in terms of Gene Ontology or pathways names (PANNZER [22], NetGO [23]).

GO functional annotation has become the standard tool in computationally based bioinformatics analyses. Due to this, the majority of method development in functional annotation is nowadays focused on GO classes, e.g., GOtcha [24], Argot [25], and Blast2GO [26]. A more comprehensive list of protein function prediction tools can be found in the review [8].

Three pipelines of function annotations (homology detection, protein–protein interaction network inference, and structure template identification) have been exploited by COFACTOR. Detailed analyses show that structure template detection based on low-resolution protein structure prediction makes a significant contribution to enhancing the sensitivity and precision of the annotation predictions, especially for cases that do not have sequence-level homologous templates [27].

## 2. Materials and Methods

### 2.1. neXtProt Versions

Jack London’s famous quote as applied to proteins might sound like this: “The proper function of a protein is to live, not to exist”. To date, 12% of the protein-coding fraction of the human genome encodes proteins with an unknown function. Additionally, suppose we adhere to the opinion about the nonwastefulness of the cell. In that case, all of these produced proteins with yet-unknown functions are here for a reason—they are doing something. 

In this study, we analyzed the terms used in neXtProt for a brief description of the protein functions to understand how and why the “profile” of the functional diversity of the human proteome has changed in recent years.

To perform the study, neXtProt was used locally. The data accessible by the resource were downloaded from the official site as a document-oriented database (DB) in XML format. The data schema was also downloaded from the project site as an XSD file. The first neXtProt release available for download (8 August 2011), the current release (beginning of 2021), and interim releases (we selected the first releases of 2012–2020) were used in the study. 

### 2.2. Instruments to Compare neXtProt Versions

The R language and some instruments for data processing and visualization of the results were used when comparing the versions of neXtProt (XSD Diagram, XMLReader, simpleXML). 

The full list of packages tested and/or used in some way is as follows:(1)heatmap: Pretty Heatmaps, Raivo Kolde, 2019, R package version 1.0.12, https://CRAN.R-project.org/package=pheatmap (accessed on 11 March 2022);(2)wordcloud2: Create Word Cloud by “htmlwidget”, Dawei Lang and Guan-tin Chien, 2018, R package version 0.2.1, https://CRAN.R-project.org/package=wordcloud2 (accessed on 11 March 2022);(3)viridis: Default Color Maps from “matplotlib”, Simon Garnier, 2018, R package version 0.5.1, https://CRAN.R-project.org/package=viridis (accessed on 11 March 2022);(4)viridisLite: Default Color Maps from “matplotlib” (Lite Version), Simon Garnier, 2018, R package version 0.3.0, https://CRAN.R-project.org/package=viridisLite (accessed on 11 March 2022);(5)gridExtra: Miscellaneous Functions for “Grid” Graphics, Baptiste Auguie, 2017, R package version 2.3, https://CRAN.R-project.org/package=gridExtra (accessed on 11 March 2022);(6)RColorBrewer: ColorBrewer Palettes, Erich Neuwirth, 2014, R package version 1.1-2, https://CRAN.R-project.org/package=RColorBrewer (accessed on 11 March 2022);(7)forcats: Tools for Working with Categorical Variables (Factors), Hadley Wickham, 2020, R package version 0.5.0, https://CRAN.R-project.org/package=forcats (accessed on 11 March 2022);(8)stringr: Simple, Consistent Wrappers for Common String Operations, Hadley Wickham, 2019, R;(9)dplyr: A Grammar of Data Manipulation, Hadley Wickham and Romain Francois and Lionel Henry and Kirill Muller, 2021, R package version 1.0.7, https://CRAN.R-project.org/package=dplyr (accessed on 11 March 2022);(10)purrr: Functional Programming Tools, Lionel Henry and Hadley Wickham, 2020, R package version 0.3.4, https://CRAN.R-project.org/package=purrr (accessed on 11 March 2022).

In the comparison, the evidence for the existence of the protein was determined based on neXtProt data for 2021. Data on proteins and their functions were initially extracted from the latest available version, and data from earlier versions were mapped to neXtProt 2021 data by a protein identifier. In doing so, proteins not represented in earlier versions were considered separately.

We selected proteins annotated with functions during the last five years at the further stage. We extracted all the publications associated with these functions and analyzed them manually. In total, we surveyed more than 600 publications. Based on their contents and keywords, we revealed the most common experimental practices for experimental protein annotation.

We chose a few examples of proteins with isoforms and analyzed the availability of their functions in automatic and manual modes.

## 3. Results

### 3.1. neXtProt Version Comparison

We analyzed neXtProt releases to evaluate the increase in the number of identified human proteins and the change in the number of proteins with known functions. When comparing the versions, only records characterized by evidence of the existence of a protein (PE1) were considered.

Most of the master proteins were known at the time of creating the first available version of neXtProt: in 2011, 17,375 proteins were known, and by 2019, there were data on another 319 proteins. The last three years have turned out to be the most fruitful in replenishing the base with new proteins, and most of the added entries relate to the products of genes of three chromosomes (2, 14, and 22—Figure 1a).

The characterized protein function and its representation in the database is not an absolute truth: we identified cases where previously discovered protein function was excluded from subsequent releases of neXtProt. A total of 351 such cases were identified. Most of the functions of the proteins were known at the time of the creation of the first available version of neXtProt; data on the function of 14,332 proteins could be obtained from neXtProt11; by 2021, the number of annotated proteins increased by a quarter, to 18,008 (Figure 1b). In total, at the beginning of 2021, for 349 proteins, the function remained unknown.

In the period 2011–2021, the establishment of the functions of known proteins was more fruitful than the identification of new proteins. For almost every chromosome, except perhaps the Y-chromosome and mitochondrial DNA, most genes and their protein products have been characterized in terms of function in neXtProt. However, these results should be interpreted with caution since, at this stage of the study, the evidence base for the establishment of function was not considered and the delay occurring between the detection of protein function in the experiment and the entry of data on it into neXtProt was not assessed.

### 3.2. Proteins Whose Functions Have Been Discovered over the Past Five Years According to Data on neXtProt

neXtProt data were used to identify proteins whose functions were first described in the last five years. Next, for these records, the types of evidence for establishing the protein function were examined. Only those records confirmed in a scientific article published from 2016 to 2021 were selected.

Over the period under review, neXtProt was replenished with data on the function of 1437 proteins for which functions were not previously known or were removed from neXtProt. 

Separate records of protein function are grouped into categories consistent with the function’s biological nature and the evidence proving its presence. For the 1437 proteins under consideration, neXtProt contains 6354 records on the presence of function; the highest number of records describe the protein as a participant in a particular biological process, while the lowest number of entries are on the presence of catalytic activity (Table 1).

Unfortunately, not all records of function in neXtProt have terms assigned to them from a controlled list (for the 1437 proteins under consideration, for the 6354 records of function, there are only 4267 with terms assigned to them). Working with records to which no terms are assigned is nontrivial. To conclude, it is necessary to generalize heterogeneous functions, which requires deep knowledge of almost everything that happens with proteins in the human body. In addition, the results of such work will be a priori controversial since the terms introduced do not belong to a controlled list approved by the scientific community. Therefore, when performing this part of the study, records of the function without controlled terms were not considered, and records with any other annotation flaws were also deleted, as a result of which 1392 proteins remained. Their annotations are visualized in Figure 2 in the form of a tag cloud on which the size of a word (function) is proportional to its frequency of occurrence. Since the frequency of occurrence of the analyzed terms varied greatly, it was scaled according to Equation (1):(1)$$\;x^{\prime }=1+\frac{19*\left(x-x_{min}\right)}{x_{max}-x_{min}}$$

An exception was made for the term “protein binding”, which was found much more often than others. We excluded it from further analysis. However, the term “protein binding” is not specific and does not contribute to understanding the protein behavior in complex systems. Protein binding is a highly generalized function description, does not provide more detailed information about the actual function of a protein, and in many cases may indicate a nonfunctional and nonspecific binding. If it is the only annotation gained by a protein, then it is hardly an advance in our understanding of that protein [28].

It is noteworthy that the detection frequency of most functions has remained unchanged throughout the existence of neXtProt. At the same time, one small subset of the protein functions began to be detected more frequently and another subset began to be detected less frequently than the average over the entire lifetime of the resource. 

Table 2 shows that the functions of proteins related to the interaction with ATP, as well as functions associated with the regulation of transcription, began to be detected less frequently. Kinases and other ATP-binding proteins, including transcriptional regulators, are important groups of proteins whose representatives are interesting for elucidating the mechanisms of development of pathological conditions and as the point of application of the therapeutic intervention. Close interest in them could be the reason for the decrease in the number of unknown representatives. 

Some functions began to be detected much more often in the last five years. They are all directly or indirectly associated with the immune response. This fact correlates well with the interests of researchers in corresponding fields of knowledge, including immuno-oncology.

Several interpretations of this fact can be assumed. Concerning functions that began to be detected less, we simply “run out” of proteins with those functions, and the interest of researchers has shifted from one type of function to another.

The current results of the analysis of protein functions are consistent with the trends observed in biological research and serve as an indirect confirmation of their fruitfulness: a change in the priority area of interest leads to the accumulation of results in a new field of research. 

A total of 1392 proteins previously selected were reviewed for available evidence of the existence of the function. For the listed proteins, 8604 pieces of evidence were extracted from neXtProt21, of which 44 pieces of evidence did not indicate the presence of a function but rather its absence. In further study, such negative evidence was not taken into account.

In addition to the data obtained during the experiments, the results of calculations (analysis of sequence similarity and its variants) and the conclusions of the curators and authors of the original publications, which are not accompanied by direct experimental confirmation, also appear as evidence.

Such data make it possible to preliminarily distinguish between the functions of proteins that can currently be used in further practical work and those that should be further investigated since solid experimental evidence is still lacking.

In total, at the current stage of the study, 3411 pieces of evidence were revealed, which are based on published experimental data.

### 3.3. Analysis of Publications Describing Experiments during Which the Functions of Proteins Were First Established in the Period from 2016 to 2021

In the framework of this study, we performed an analysis of publications describing experiments in which the functions of proteins were first discovered in the period from 2016 to 2021. Experimental evidence and the corresponding publications (353 articles from 2016 to 2021) were found only for 708 proteins:On average, a publication confirms the functions of 3.81 proteins.In 135 publications, data are provided on the function of only one (but not the same, of course) protein.On average, there are 1.5 publications per protein with confirmation of its functions.Overall, 611 proteins have a confirmation of function in only one publication.In all, 58% of all the functions shown were binding to other proteins (“protein binding”).

In the final step, a comparison was made of the dates of the detection of the function of binding other proteins and other functions. A summary of the results of the comparison is shown in Table 3.

It follows from Table 3 that, at least in the last five years, the determination of protein binding is essential for determining other protein functions. For 58% of proteins with known functions, in addition to any function, binding is also shown. Of these, only nine proteins (5% of proteins with known function, in addition to protein binding) were annotated by any function before the binding was shown. Thus, in many cases, binding determinations are the first step in functional protein annotation. The estimates obtained in this study are approximate since they do not consider the hierarchy of terms describing the functions of proteins. Among the functions of proteins, some are affiliated with the binding; i.e., the above estimates may underestimate the significance of binding by determining other functions.

In Table 3, it is also worth paying attention to the fact that in the analyzed articles for most of the proteins, only binding was determined as a function (72% of all proteins). This may lead to the fact that there is an extensive field for clarifying the functions of proteins for which their high-molecular-weight partners are known. Such work ends with conclusions that do not directly follow experiments (such evidence and functions were not considered).

The study results show that over the past five years, more than 600 proteins have received functional annotation, many of which have been assigned several functions. Of particular interest are the studies and the methods used in them, in which the presence of function was determined for the protein for the first time. Such publications were identified as follows:The publication dates for the candidate articles under consideration were extracted from neXtProt and, if necessary, supplemented (in some cases, only the year was indicated for publication, which may not be enough to determine the primacy).The list of publications, evidence, annotations, and proteins was sorted by publication date and protein identifier.Multiple records of protein function were deleted, except for the one relying on the oldest publication for the period under review.In this way, records were deleted regarding all the functions, except for one. Several protein functions could be determined; the missing data were then reattached by the coincidence of the identifiers of the protein and publication.

Moreover, in one publication [29], the functions of 379 proteins were first established (58% of all proteins for which functions were established in experiments and reliably described in the period from 2016 to 2021). It makes sense to describe this study in more detail.

The authors of the study, relying on data from proteomic studies, selected pairs of proteins previously not shown experimentally as interacting and, using the two-hybrid yeast system, tested the presence of interaction between them. This method is based on the ability of two domains of the yeast protein GAL4 (transcription activator) to maintain functional activity in the composition of chimeric proteins, which, due to the interaction between chimeric proteins, exhibit GAL4 activity and activate the transcription of the reporter gene, confirming the presence of an interaction between proteins of which they are fragments. The authors validated the results using alternative methods for determining protein–protein interactions. Thanks to the efforts made by the authors to select pairs of proteins for testing interactions, they validated the method and statistically processed the results using a relatively simple procedure. They succeeded in determining most of the protein functions first discovered in the last five years.

The next article [30] (by the number of proteins, the functions of which were described for the first time) revealed the functions for eight proteins for the first time.

In this study, the authors identified partners using methods relating to protein–protein interaction:AP-MS: affinity purification of labeled (3xFLAG) and interacting proteins with subsequent mass spectrometric identification of the components of the complex.LUMIER: a method based on copurification and luminescence of complexes: the gene of one of the potential participants of the complex was fused with a sequence encoding luciferase and stably expressed in 293T cell lines; genes of other potential partners with 3xFLAG incorporation were transfected into cells of the reporter line. Cell lysates were incubated together in the presence of antibodies for the label (3xFLAG), and after washing, the formation of a stable complex was determined by the presence of luminescence.

Some of the complexes were determined by both methods, while some of the complexes were known from the literature. Thus, the AP-MS method was validated as part of the objective and showed high reliability in the determination of complexes after appropriate statistical processing of raw data.

The next article [31] (by the number of proteins, the functions of which were described for the first time) revealed the functions for six proteins for the first time. Interestingly, the functions were also defined for proteins that underwent post-translational modification, including phosphorylation at tyrosine residues. The authors used a modified two-hybrid yeast system to identify protein–protein interactions. In the classical version, this method does not allow for detecting protein–protein interactions that depend on those post-translational modifications that do not proceed normally in yeast cells. The authors introduced into yeast cells an additional plasmid, which contained the human non-receptor tyrosine kinase gene, the product of which carried out the phosphorylation necessary to evaluate the interaction between the modified proteins. The presence of interaction was defined as the growth of yeast colonies since the beta-galactosidase gene required for growth under the cultivation conditions was used as the reporter gene, the expression of which depended on the presence of interaction. As a direct interaction partner for phosphorylated proteins, proteins capable of recognizing phosphorylated tyrosine residues due to the presence of the SH2 or PTB domain were used. This method was validated using previously established protein–protein interactions. Differentiation between interactions that require and do not require phosphorylation was carried out because some experiments were performed using protein kinases lacking enzymatic activity. Thus, the authors established some proteins that undergo phosphorylation at tyrosine residues and partners for their further interaction.

Based on the analysis of individual publications, it seems that the primary approach that is currently used to identify the functions of proteins is to determine the partners of their interactions in the cell either by phenotypic screening (a two-yeast system, for example) or by mass spectrometric analysis of the contents of the cell; subsequent analysis of data on multiple protein partners allows us to move from binding data to substantiated hypotheses about the role of the protein in the life of the cell as a whole and in individual events.

## 4. Discussion

The above shows that neXtProt is a powerful source of protein function information, corresponding to current guidelines [32,33,34,35,36,37]. We need to mention that it is easy to obtain any existing information about proteins and their isoforms in manual mode. Otherwise, the current data frame is not optimized for the proteoform-centric mode in the context of proteoform functions. At the moment, neXtProt reports additional splice isoforms for half the human proteome (10,535 entries). For 916 of these entries, it was shown that the different splice isoforms have a different subcellular location or function [38]. Of course, the evolution of computational technologies makes it possible to predict protein functions with a high degree of reliability and perform GO annotations based on protein sequences and structures. However, we observe that even advanced bioinformatic algorithms face the problem of functional annotation of noncanonical protein variants, and curated repositories, to the same extent, face issues regarding the efficient storage of information about these proteoforms.

It is well known that alternatively spliced proteoforms of one gene can have different, sometimes opposed functions, as happened in the case of two splice variants of the Bcl-x gene. A long version Bcl-xL protein inhibits programmed cell death; it is antiapoptotic. Alternatively, short form Bcl-xS antagonizes the inhibitory functions of Bcl-xL. We observe a similar situation with the example of point amino acid substitutions. There is a vivid example of the “butterfly effect” of functional heterogeneity produced by an aberrant protein PIK3CA:H1047R. This protein (its canonical variant) is responsible for cell growth, proliferation, and survival in healthy cell line MCF-10A. Single amino acid H1047R substitution induces the transformation of the expression profile of a healthy cell into a profile typical of a malignant tumor. Extensive cellular reorganization associated with this substitution far exceeds basic activities of PI3K and affects structural protein levels, the DNA repair machinery, and metabolism.

These and many other cases prompted us to think about the importance of finding out the function for each proteoform and using reliable tools for these tasks, including bioinformatics and classical laboratory ones.

## 5. Conclusions

Comparing versions of neXtProt demonstrates that annotating proteins with new functions is a more fruitful process than the detection of “missing” proteins due to the domination of bioinformatical approaches, which are less expensive than the experimental methods required to validate the existence of “missing” proteins. The proportion of proteins with known functions is higher for medium and small chromosomes. The most commonly determined protein functions change little over time: the binding protein function is often determined. However, in the last five years, protein functions associated with immunity have become much more common to discover than functions associated with transcriptional regulation and phosphorylation. The publication of information on protein function most often begins with data on the protein binding to other proteins. The methods that make the most significant contribution to identifying protein–protein interactions are various options for mass spectrometry and the use of a two-hybrid yeast system.

## Figures and Tables

**Figure 1 jpm-12-00479-f001:**
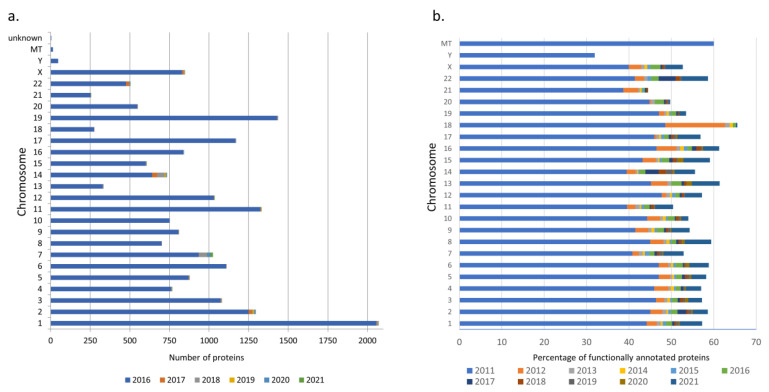
Change in the completeness of human protein data according to neXtProt. (**a**) Chronology of changes in the number of protein identifications; (**b**) chronology of replenishment of neXtProt with information on protein functions.

**Figure 2 jpm-12-00479-f002:**
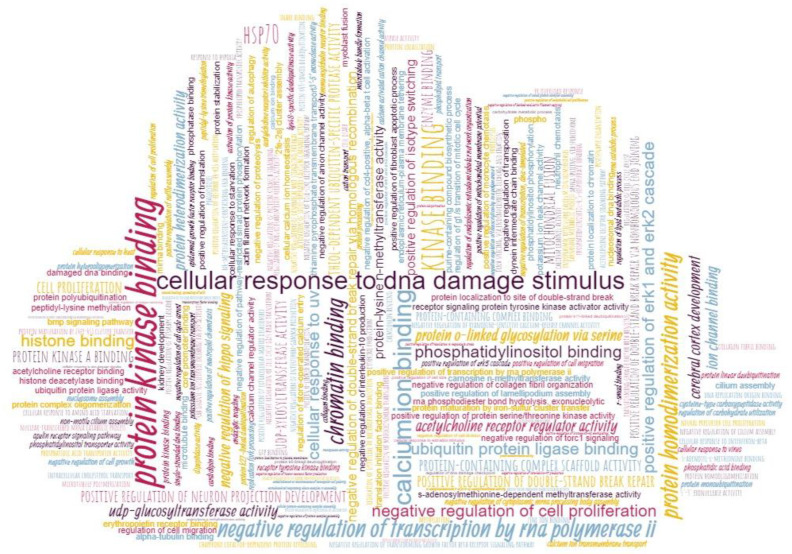
A cloud of biological functions for proteins. The data presented refer to 1392 proteins whose functional annotation appeared in neXtProt from the beginning of 2016 to the beginning of 2021.

**Table 1 jpm-12-00479-t001:** Categories of functional annotation and the number of records for 1441 proteins whose functions were first annotated in one of the versions of neXtProt over the past five years.

Categories	Number of Records
catalytic-activity	39
function-info	802
go-biological-process	2684
go-molecular-function	1571
Pathway	1303
transport-activity	98

**Table 2 jpm-12-00479-t002:** Functions of proteins that began to be detected more/less frequently according to neXtProt over the past five years.

Functions Detected More Frequently	Functions Detected Less Frequently
antigen binding	ATP binding
immunoglobulin production	DNA-binding transcription factor activity, RNA polymerase II-specific
immunoglobulin receptor binding	DNA binding
phagocytosis, recognition	DNA-binding transcription factor activity
positive regulation of B cell activation	regulation of transcription by RNA polymerase II
phagocytosis, engulfment	positive regulation of transcription, DNA-templated
complement activation, classical pathway	oxidation–reduction process
B cell receptor signaling pathway	positive regulation of transcription by RNA polymerase II
immune response	protein serine/threonine kinase activity
defense response to bacterium	regulation of transcription, DNA-templated
adaptive immune response	

**Table 3 jpm-12-00479-t003:** The sequence of identification of protein functions related to binding to other proteins is estimated according to neXtProt using a selection of identified functions and evidence such as direct assay evidence used in manual assertion and physical interaction evidence used in manual assertion, referencing journal articles from 2016 to 2021.

The Sequence of Identification of Protein Functions	Number of Cases
Protein binding and any other function have been identified in one publication	79
Protein binding has been shown prior to determining any other function	58
Protein binding has been shown after determining any other function	14
Protein binding has not been shown, but other function has been defined	73
To date, only protein binding has been shown	442

## Data Availability

Not applicable.

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
