# Peer review of "Evolution of Protein Functional Annotation: Text Mining Study"

_jpm, 2022, doi:10.3390/jpm12030479_

Round 1
Reviewer 1 Report
In this manuscript, the authors introduced that the neXt-CP50 Challenge that is launched under the framework of HPP. It faced to the challenge that the concept of protein function, specific experimental and bioinformatics protocols are not standardized. They proposed the NextProt database to tackle these issues, by identifying the most frequently used experimental and bioinformatic methods for analyzing protein functions.
The authors concluded that annotating proteins with new functions is more fruitful process than ”missing” proteins detection due to the domination of bioinformatical approaches. They also demonstrated that most commonly determined protein functions change little over time, and the most often binding protein function is determined.
I consider the authors comprehensively demonstrate the novelty and necessity of this work.
Author Response
In this manuscript, the authors introduced that the neXt-CP50 Challenge that is launched under the framework of HPP. It faced to the challenge that the concept of protein function, specific experimental and bioinformatics protocols are not standardized. They proposed the NextProt database to tackle these issues, by identifying the most frequently used experimental and bioinformatic methods for analyzing protein functions.
The authors concluded that annotating proteins with new functions is more fruitful process than ”missing” proteins detection due to the domination of bioinformatical approaches. They also demonstrated that most commonly determined protein functions change little over time, and the most often binding protein function is determined.
I consider the authors comprehensively demonstrate the novelty and necessity of this work.
Answer: Dear Reviewer, thank you so much.

Reviewer 2 Report
The authors have presented a study understanding the evolution of annotating protein function by teasing the neXtProt database over a period of time. Here are the a few suggestions to further improve the manuscript -
- Can authors detail the process used to generate the protein function cloud?
- In the formula used to scale/normalize the frequency of occurrence, how did the authors chose the factor '19'? What does it represent? Can the authors elaborate that?
- In methods section, authors mention that some instruments were used for data processing and visualization. Can they elaborate specifically what instruments were used?
- Did the authors limit their search to the neXtProt database or did they consider other databases as well such as UniProt, InterPro, etc.?
Author Response
Can authors detail the process used to generate the protein function cloud?
Answer: Thank you for your interest, initially wordCloud2 package was used to generate word cloud as follows:
library(wordcloud2)
palFun <- c("#2a4f8e", "#4DBBD5FF", "#3a745f", "#3a4f74", "#b1576f", "#8491B4FF", "#2a8e69", "#E54B4B", "#7E6148FF")
data <- read.delim("C:\\...\\nextProt\\forReprt\\data\\uCV.tab")
wordcloud2(data = data, size = 0.8, color=sample(palFun, size = 100, replace = TRUE), minRotation = pi/2, maxRotation = pi/2)
In the formula used to scale/normalize the frequency of occurrence, how did the authors chose the factor '19'? What does it represent? Can the authors elaborate that?
Answer: Thank you for your question. Factor value was chosen to preserve differences between words in terms of their occurrence and, at the same time, make them readable (as many readble words as possible). It was trial and error process.
In methods section, authors mention that some instruments were used for data processing and visualization. Can they elaborate specifically what instruments were used?
Answer: Yes, here is the full list of packages, which somehow had been tested and/or used:
- heatmap: Pretty Heatmaps,Raivo Kolde,2019,R package version 1.0.12,https://CRAN.R-project.org/package=pheatmap,
- wordcloud2: Create Word Cloud by 'htmlwidget',Dawei Lang and Guan-tin Chien,2018,R package version 0.2.1,https://CRAN.R-project.org/package=wordcloud2,
- viridis: Default Color Maps from 'matplotlib',Simon Garnier,2018,R package version 0.5.1,https://CRAN.R-project.org/package=viridis,
- viridisLite: Default Color Maps from 'matplotlib' (Lite Version),Simon Garnier,2018,R package version 0.3.0,https://CRAN.R-project.org/package=viridisLite,
- gridExtra: Miscellaneous Functions for "Grid" Graphics,Baptiste Auguie,2017,R package version 2.3,https://CRAN.R-project.org/package=gridExtra,
- RColorBrewer: ColorBrewer Palettes,Erich Neuwirth,2014,R package version 1.1-2,https://CRAN.R-project.org/package=RColorBrewer,
- forcats: Tools for Working with Categorical Variables (Factors),Hadley Wickham,2020,R package version 0.5.0,https://CRAN.R-project.org/package=forcats,
- stringr: Simple, Consistent Wrappers for Common String Operations,Hadley Wickham,2019,R
- dplyr: A Grammar of Data Manipulation,Hadley Wickham and Romain Francois and Lionel Henry and Kirill Muller,2021,R package version 1.0.7,https://CRAN.R-project.org/package=dplyr,
- purrr: Functional Programming Tools,Lionel Henry and Hadley Wickham,2020,R package version 0.3.4,https://CRAN.R-project.org/package=purrr.
Did the authors limit their search to the neXtProt database or did they consider other databases as well such as UniProt, InterPro, etc.?
Answer: Thank you for your comment, we worked with the neXtProt, which integrates data from many other sources including UniProt and InterPro. The development of neXtProt is carried out in close collaboration with the Swiss-Prot group and the data in the neXtProt and the Uniprot can differ, but not dramatically. The NextProt focuses on human proteome, whether the UniProt includes data on the great number of organisms. Our work was limited by the human proteome, but the same approach can be used for any other organism, based on the Uniprot data in the future research.
Reviewer 3 Report
In this article, the authors were interested in studying protein functions through bioinformatics databases such as the neXtProt database.
This kind of article should be written and published to make the knowledge of "-omics", accessible to all scientists without computer bases.
Just a minor comments, why the authors did not use or refer at all to a comparison with the UNIPROT databases.
Author Response
Thank you for your comment, we worked with the neXtProt, which integrates data from many other sources including UniProt and InterPro. The development of neXtProt is carried out in close collaboration with the Swiss-Prot group and the data in the neXtProt and the Uniprot can differ, but not dramatically. The NextProt focuses on human proteome, whether the UniProt includes data on the great number of organisms. Our work was limited by the human proteome, but the same approach can be used for any other organism, based on the Uniprot data in the future research.